# Prevention of Breast Cancer-Related Lymphedema: An Up-to-Date Systematic Review of Different Surgical Approaches

**DOI:** 10.3390/jcm13020555

**Published:** 2024-01-18

**Authors:** Domenico Pagliara, Federica Grieco, Silvia Rampazzo, Nicola Pili, Pietro Luciano Serra, Roberto Cuomo, Corrado Rubino

**Affiliations:** 1Plastic-Reconstructive and Lymphedema Microsurgery Center, Mater Olbia Hospital, 07026 Olbia, Italy; domenico.pagliara@materolbia.com; 2Plastic Surgery Unit, University Hospital Trust of Sassari, 07100 Sassari, Italy; s.rampazzo@studenti.uniss.it (S.R.); n.pili2@studenti.uniss.it (N.P.); p.serra4@studenti.uniss.it (P.L.S.); corubino@uniss.it (C.R.); 3Department of Medicine, Surgery and Neurosciences, University of Siena, 53100 Siena, Italy; roberto.cuomo@unisi.it

**Keywords:** breast cancer-related lymphedema, lymphatic surgery, microsurgery, primary prevention, lymphedema

## Abstract

Breast cancer-related lymphedema (BCRL) affects approximately 20% of women after breast cancer therapy. Advances in treatment have increased the life expectancy; thus, the prevalence of BCRL will continue to rise with the number of cancer survivors, hence the need to develop strategies to prevent this condition. We provide a systematic review of the literature on the primary prevention of BCRL by prophylactic lymphatic surgery (PLS). Between June and August 2022, we conducted a search of PubMed, Google Scholar and Cochrane. In the end, a total of eighteen papers were selected. The eleven studies without a control group reported only 15 of 342 patients who developed lymphedema at least six months after PLS (4.59%). The seven studies with a control group included 569 patients, 328 cases and 241 controls. Among the cases, 36 (10%) developed lymphedema. In contrast, the incidence of lymphedema in the controls was 40% (98 of 241 patients). The formulation of definite recommendations in favor of PLS is hindered by low-quality studies. There is no consensus on which technique should be preferred, nor on whether adjuvant radiotherapy might affect the efficacy of PLS. Randomized controlled trials are mandatory to conceive evidence-based recommendations.

## 1. Introduction

Lymphedema is characterized by cumulative tissue swelling caused by the impaired drainage function of the lymphatic vessels. It may be the result of aberrant lymphatic development, as in primary cases, or be secondary to traumatic or iatrogenic injury to lymph nodes or lymphatic vessels [1].

The condition is both physically and psychologically distressing, as patients suffer from chronic pain and the decreased strength and function of the affected limbs [2].

In the Western world, lymphedema is most commonly associated with secondary cases related to breast cancer treatment [3].

The incidence of BCRL after axillary lymph node dissection (ALND) varies widely, ranging from 14.1% to 33.4%, with the highest rates reported in patients undergoing adjuvant radiotherapy [4]. Several conservative strategies have been implemented to reduce the risk of developing lymphedema in breast cancer patients after ALND, or to treat it once it has occurred [5,6].

The most commonly used options include exercise, manual lymph drainage, compression therapy and lymph taping (kinesio tape). Although there is currently no consensus on the most effective conservative treatment option or combination of options, surgery has emerged as a “last-ditch” effort when lymphedema reoccurs after all other measures have failed [7].

Surgical techniques currently utilized in the treatment of lymphedema include liposuction, vascularized lymph node transfer (VLNT) and lymphatic–venous anastomosis (LVA) [8].

The former directly aims to reduce the volume of the affected limb, whereas VLNT affects lymphatic drainage by potentially inducing the formation of lymphatic vessels over time. LVA, on the other hand, diverts lymphatics directly into the venous circulation of the arm, bypassing the impaired lymphatic drainage. The clinical efficacy of the above options varies greatly, whether conservative or surgical, with most findings reporting only marginal improvements or none whatsoever [9]. In 2010, prophylactic LVA following ALND was conceived as a strategy for the primary prevention of lymphedema, under the protocol known as the Lymphatic Microsurgical Preventing Healing Approach (LYMPHA) [3,10].

Few studies have quantitatively assessed the impact and clinical outcomes of this intervention in the primary prevention of BCRL. This manuscript aims to present a systematic review of the state of the art and the most current evidence supporting the use of PLS.

## 2. Materials and Methods

### 2.1. Literature Search

A review of the literature was carried out according to the Preferred Reporting Items for Systematic Reviews and Meta-Analyses (PRISMA) statement. Registration of the review on the international database of prospectively registered systematic reviews PROSPERO was not performed.An electronic search was conducted through PubMed/Medline, Google Scholar and Cochrane Database between June and August 2022 by two independent reviewers (DP and FG). The databases were searched using the following Medical Subject Headings (MeSH) terms “lymphatic venous anastomosis”; “lymphaticovenular anastomosis”; “breast-cancer-related-lymphedema”; “lymphaticovenular bypass”; “prophylaxis”; “vascularized lymph node transplant”; “lymphovenous bypass”, in combination with AND or OR. A manual search of references was also conducted to identify any other potentially relevant additional studies. Inclusion criteria for manuscripts, used for full-text assessment and data extraction, were English-based original papers exclusively discussing PLS and, specifically, randomized clinical trials, retrospective comparative studies, retrospective case series of at least three patients and prospective studies. Studies addressing lymphatic mapping alone or single case reports were discarded. Prophylaxis was defined as therapeutic interventions directed towards the prevention of disease from occurring; hence, manuscripts discussing subjects with clinical or subclinical evidence of lymphedema were excluded.

### 2.2. Data Extraction

Data extraction was performed by the same two independent reviewers (DP and FG) who reported the extracted data in a spreadsheet where all the relevant information was included. Any case of divergent opinions was solved either through consensus-based discussion or through the intervention of a third independent investigator (CR) and a majority-based vote. For all included studies, the following data were documented: type of study, year of publication, authors, number of patients, type of microsurgical technique, operating time, follow-up period, complications, presence of a control group, method of lymphedema diagnosis, lymphedema outcome, whether patients received adjuvant therapy and follow-up time. The Oxford Centre for Evidence-Based Medicine (OCEBM) and the JADAD scale were used to provide a critical appraisal of the level of evidence.

## 3. Results

### 3.1. Study Characteristic

A total of 5028 articles were identified in the Cochrane, Google scholar and Pubmed search. After the removal of duplicates, the literature search retrieved 2698 studies, of which only 55 full-text papers were assessed for eligibility. Six abstracts, ten overviews, three systematic reviews, one meta-analysis and one case report were excluded, identifying 34 eligible full-text articles. After the full-text screening, an additional 16 articles were eliminated either because of population overlap, or due to the management of upper or lower lymphedema already diagnosed or because of no surgical prophylactic techniques. In the end, 18 studies were included (Figure 1).

An overview of each study is summarized in Table 1 and Table 2. Nine studies were observational cohort studies, five were case–control, two were randomized trials and two were case series.

The OCEBM level of evidence for the articles was as follows: eleven articles had a level of 4, five a level of 3 and two a level of 2. The JADAD level score was 5 for two articles, 4 for six articles and 3 for ten articles.

### 3.2. Patient Characteristics

The qualitative analysis comprised 911 patients who had received ALND for breast cancer treatment. Among the overall eighteen studies, eleven had no control group, while seven had a control group. The eleven studies without a control group reported only 15 patients out of 327 who developed lymphedema (4.59%) after at least six months from PLS. Of these, eight had received radiotherapy exclusively, one had received adjuvant radiotherapy and neo-adjuvant chemotherapy and had a body mass index (BMI) of 38, one had received radiotherapy and had a concomitant axillary surgical site infection and seven had received neo-adjuvant chemotherapy and adjuvant radiotherapy [11,12,13,14,15]. The seven studies with a control group comprised 569 patients, of which 328 were cases and 241 were controls. Among the cases, 36 (10%) developed lymphedema. These patients had either received adjuvant radiotherapy, had an increased number of positive nodes removed, or had been treated with adjuvant chemotherapy and radiotherapy and had a BMI > 40 [16,17,18,19]. Among the controls, instead, the rate of lymphedema was 40% (98 patients out of 241). All the lymphedemas diagnosed within the six months of their final oncologic treatment (chemotherapy, radiotherapy and surgery) were defined as transient and thus not taken into account.

The follow-up period ranged from three to eighty-four months, during which, besides lymphedema, only one complication was registered: an infected axillary seroma treated with aspiration and intravenous antibiotics [12].

The methods used to detect lymphedema differed among studies. Fourteen studies applied circumferential arm measurements (CA) [12,13,14,15,18,19,20,21,22,23,24,25,26,27]. Four papers used volumetry (V) and calculated the relative volume change by dividing the pre- and postoperative difference by the preoperative value [11,16,17,28]. The estimated limb volume was calculated using the following formula: V = (d)(A2 + Aa + a2)/12(π), where “A” is the circumference measurement of the distal section, “a” is the circumference measurement of the proximal section, and “d” is the distance between the distal and the proximal section [29]. Nevertheless, seven studies also used, but not exclusively, bioimpedance spectroscopy (BS) [12,13,18,19,24,25,26,27]. Some studies also used lymphoscintigraphy as a tool to diagnose lymphedema, in addition to other methods [11,15,20]. Only one study applied the Lymphedema Life Impact Score and/or Lymphedema Quality of Life Questionnaire carried out by a certified lymphedema therapist [18].

Almost all the studies performed LVA as primary prophylaxis isotopic to the lymphadenectomy procedure. Telescopic end-to-end anastomosis of several lymphatics inserted together into a single vein was reported in ten studies [11,14,15,16,17,18,20,21,24,27]. Two studies opted for super microsurgical LVA exclusively and two studies for a combination of both techniques [12,13,24,26]. Two studies used both an end-to-end LVA and a telescopic end-to-side technique of multiple lymphatics on the same vein [19,23]. Only one study performed multiple distal LVA between the proximal end of the vein and the distal end of the lymphatic duct, as a primary prophylaxis ectopic to the lymphadenectomy procedure [24]. One paper described autologous breast reconstruction using the deep inferior epigastric perforator (DIEP) flap and the lymphatic superficial circumflex iliac artery perforator (SCIP) flap procured separately from the Zone 4 region [28]. Another paper accounted for a vascularized serratus anterior fascia flap during concurrent latissimus dorsi flap harvest (for breast or chest wall reconstruction) [22]. In terms of feasibility, all the techniques reported were highly feasible, with a pooled feasibility rate varying from 75% up to 100%.

**Table 1 jcm-13-00555-t001:** Studies with a control group.

Article, Year of Publication and Number of Reference	Adjuvant Therapy	LVA Shunting Technique	LVA Feasibility	Follow-Up (Months)	Operating Time (min)	Method of Lymphedema Diagnosis	Cases with Lymphedema	Controls with Lymphedema	Cases with Lymphedema Who Received Adjuvant Radiotherapy	OCEBM and JADAD
Boccardo 2011 [16]	Cases: RT (11/23) Controls: RT (12/23)	Sleeve	23/23 (100%)	18	15–20	V	1/23 (4%)	7/23 (30%)	1/1 (100%)	2 and 5
Feldman 2015 [17]	Cases: RT (15/24), CT (23/24) Controls: RT (6/8), CT (7/8)	Sleeve	24/32 (75%)	From 3 to 24	45	V	3/24 (13%)	4/8 (50%)	3/3 (100%)	3 and 3
Hahamoff 2019 [25]	Cases: RT (8/8), neoCT (5/8), adCT (4/8); Controls: RT (6/10), neoCT (4/10), adCT (4/10)	Sleeve	8/8 (100%)	From 15 to 20	From 32 to 95	CA, BS	0/8 (0%)	4/10 (40%)	0/0	3 and 3
Herremans 2021 [18]	Cases: RT (67/76), CT (58/76), neoCT (36/76) Controls: RT (50/56), CT (42/56), neoCT (20/56)	Sleeve	76/84 (90%)	60	nr	CA, BS, LQOLQ	10/76 (13.2%)	16/56 (28.6%)	Nr	3 and 4
Yoon 2021 [26]	Cases: RT (17/21), CT (16/21); Controls: RT (36/48), CT (38/48)	ETE LVA	21/21 (100%)	6	From 30 to 60	CA, BS	0/21 (0%)	9/48 (18.8%)	0/0	2 and 5
Ozmen 2022 [27]	Cases: RT (89/110); Controls: RT (68/84)	Sleeve	Nr	From 10 to 84	nr	CA, BS	18/110 (16%)	57/84 (68%)	Nr	3 and 4
Weinstein 2022 [19]	Cases: RT (46/66), neoCT (56/66) adCT (26/66); Controls: RT (8/12), neoCT (8/12), adCT (8/12)	ETE or ETS LVA	Nr	8 on average	nr	CA, BS	4/66 (6%)	1/12 (8%)	3/4 (75%)	3 and 4

RT: radiotherapy, CT: chemotherapy, neoCT: neoadjuvant chemotherapy, adCT: adjuvant chemotherapy, ETE LVA: end-to-end lymphatic–venous anastomosis, ETS LVA: end-to-side lymphatic–venous anastomosis, nr: not reported, V: volumetry, CA: circumferential arm measurements, BS: bioimpedance spectrometry, LQOLQ: Lymphedema Quality of Life Questionnaire.

**Table 2 jcm-13-00555-t002:** Studies without a control group.

Article, Year of Publication and Number of Reference	Adjuvant Therapy	LVA Shunting Technique	LVA Feasibility	Follow-Up (Months)	Operating Time (min)	Method of Lymphedema Diagnosis	Cases with Lymphedema	Cases with Lymphedema Who Received Adjuvant Radiotherapy	OCEBM and JADAD
Boccardo 2009 [20]	RT (7/18)	Sleeve	18/19 (95%)	12	15	CA, LS	0/18 (0%)	0/0	4 and 3
Casabona 2009 [21]	RT (8/8), CT (0/8)	Sleeve	8/9 (89%)	9	17	CA	0/8 (0%)	0/0	4 and 3
Boccardo 2015 [11]	RT (35/74)	Sleeve	74/78 (95%)	48	48	V, LS	3/74 (4%)	3/3	4 and 3
Johnson 2019 [14]	RT (26/32), CT (19/32)	Sleeve	nr	12	Nr	CA, BS	1/32 (3.1%)	1/32	4 and 4
Scharwz 2019 [12]	RT (52/58), neoCT (43/58), adCT (10/58)	37/58 ETE LVA, 21/58 sleeve	58/60 (97%)	29	95	CA, BS	2/43 (4.6%)	2/2	4 and 3
Cook 2021 [15]	RT (22/33), neoCT (24/33)	Sleeve	33/33 (100%)	12	Nr	CA, LS	3/33 (9%)	3/3	4 and 4
Shaffer 2020 [13]	RT (82/88), neoCT (61/88), adCT (20/88), neo + adCT (1/88)	ETE LVA or sleeve	88/88 (100%)	14.6 on average	From 161 to 253	CA, BS	5/88 (6%)	4/5	4 and 4
Han 2022 [22]	RT (3/3), neoCT (2/3)	Vascularized serratus anterior fascia flap	nr	48	Nr	CA	0/3 (0%)	0/0	4 and 3
Lipman 2022 [23]	RT (16/19)	ETE or ETS LVA	nr	10 on average	From 32 to 95	CA, BS	1/19 (5%)	nr	4 and 3
Pierazzi 2022 [24]	RT (5/5)	DLVA	5/5 (100%)	12	Nr	CA	0/5 (0%)	0/0	4 and 3
Yoshimatsu 2022 [28]	RT (2/4)	SCIP flap with DIEP	4/4 (100%)	From 24 to 48	Nr	V	0/4 (0%)	0/0	4 and 3

RT: radiotherapy, CT: chemotherapy, neoCT: neoadjuvant chemotherapy, adCT: adjuvant chemotherapy, ETE LVA: end-to-end lymphatic–venous anastomosis, ETS LVA: end-to-side lymphatic–venous anastomosis, DLVA: distal lymphatic–venous anastomosis, nr: not reported, V: volumetry, CA: circumferential arm measurements, LS: lymphoscintigraphy, BS: bioimpedance spectrometry.

## 4. Discussion

Considering the high morbidity of ALND in breast cancer patients, several techniques attempting to reduce the lymphedema rate have been implemented over recent decades.

In 2007, Thompson and Nos described, in two different studies, the axillary reverse mapping (ARM) technique, demonstrating that arm and breast lymphatic drainages can be identified separately [30,31]. They proposed, a few minutes before proceeding with sentinel lymph node biopsy (SLNB) or ALND, the injection of a colorant (blue dye) into the upper arm to make visible during the dissection the lymphatics draining exclusively the arm, and not the breast, and preserve them. Thompson and Nos showed that their technique was feasible, with a detection rate of blue lymphatics of 61–71% and a preservation rate of 47%. After introducing ARM, the incidence of upper extremity lymphedema went from 33.4% to 4% [14,32,33,34]. Unfortunately, the ARM technique couldn’t guarantee an oncological radicality, since blue lymph nodes were considered part of the arm lymphatic pathway, and thus were not originally removed, not knowing if they were metastatic or not. Another point of controversy was the removal of the lymphatics departing from the blue nodes when exiting the axillary basin and joining the common lymphatic pathway draining the breast. According to Boneti et al., their preservation was considered not safe in terms of oncological radicality [35]. Therefore, aiming to find a technique able to prevent secondary arm lymphedema and, at the same time, maintain the oncological radicality, Boccardo et al. developed the lymphatic microsurgical preventing healing approach (LYMPHA) [20]. The microsurgical operation, also known as the “sleeve technique”, consisted of a telescopic end-to-end anastomosis: blue lymphatics found at the lateral pillar of the axillary dissection (AD) after the blue dye injection were placed together into the vein with a U-shaped stitch. The lymphatics were then stabilized inside the vein with additional stitches between the vein border and the perilymphatic tissue. As a matter of fact, Boccardo implemented the ARM technique, not saving the blue nodes and the lymphatics coming from them, and adding the LYMPHA procedure, counting zero cases of lymphedema within 12 months in an 18-patient population [20]. After Boccardo et al., Casabona also applied the ARM and the LYMPHA technique, reporting no cases of lymphedema in eight patients in a 9-month follow-up [21]. In 2015, Boccardo extended the use of LYMPHA and ARM to 74 patients of which only 3 developed lymphedema within 48 months of follow-up (4%) [11]. An inferior rate was reported by Johnson: out of 32 patients treated, only 1 developed lymphedema (3.1%) within 12 months [14]. Applying the same procedure and in the same time frame, Cook registered a 9% rate of lymphedema (3 out of 33 patients) [15].

In their study, Scharwz et al. tried to prevent lymphedema occurrence in 58 patients by applying an end-to-end micro anastomosis between a tributary of the lateral thoracic vein or the thoracodorsal vein and a single transected lymphatic, in the instance of its precise size match and availability [12]. When a significant size discrepancy existed between the lymphatic and recipient vein (1:3), or if there were multiple transected lymphatics in proximity to a recipient’s vein, they utilized the sleeve technique already described by Boccardo [20]. Unfortunately, the surgical procedure used for the two patients who developed lymphedema reported in the study was not registered; thus, it was not possible to detect the more effective type of anastomosis. In 2020, Shaffer et al. applied the same scheme as Scharwz to 88 patients with a rate of lymphedema of 6% (5 out of 88) and also in this case it was not possible to detect the relation between the type of anastomosis and the onset of lymphedema [12,13].

In 2021, Chuan et al., in three patients with locally advanced breast cancer requiring mastectomy and axillary clearance, harvested a vascularized serratus anterior fascia flap during concurrent latissimus dorsi flap dissection (for breast or chest wall reconstruction) and then wrapped it around the axillary vessels [22]. In this way, they provided a conduit for lymphatic regeneration, protecting the axillary vessels from radiotherapy and reducing scarring and axillary cording. Within 48 months, none of the patients experienced upper limb lymphedema or cording. A similar concept was applied by Yoshimatsu et al. in 2022 [28].

They detailed an innovative technique where the afferent lymphatic vessels, along with their associated lymph nodes from the Zone 4 region, were extracted as an independent flap known as the superficial circumflex iliac artery perforator (SCIP) flap. This approach was applied within the framework of autologous breast reconstruction using the deep inferior epigastric artery perforator (DIEP) flap.

In 2022, Lipman et al. applied the LYMPHA procedure to 19 patients using indistinctly end-to-end or end-to-side LVA, and had only one case of lymphedema in an average of a 10-month follow-up period. Moreover, they reported about one patient who simultaneously underwent immediate breast reconstruction with an omental-free flap [23]. As a result, in this case, it may be difficult to determine the relative contributions of LYMPHA versus omental transfer on lymphedema prevention. In fact, in the intra-abdominal space, the omentum is known to serve a critical role in immune response and lymphatic drainage. Though the omental transfer for breast reconstruction did not involve the transfer of the gastric nodal basin or lymphovenous anastomosis of the efferent lymphatic that sometimes accompanies the gastroepiploic vessels, the omentum-associated lymph tissue (OALT) within the flap may have contributed partially to improve lymphatic drainage postoperatively [36].

Finally, in 2022, Pierazzi et al. evaluated five patients who underwent prophylactic LVA distally to the axillary region and after the conclusion of adjuvant radiotherapy [24]. For each patient, the microsurgical technique was the same standard technique for the LVA procedure and four anastomoses were performed between the proximal end of a subdermal vein and the distal end of a lymphatic duct. None of them developed lymphedema within 12 months [37].

However, the real proof of the LYMPHA technique is manifested through case–control studies. Two randomized case–control papers showed a rate of 30% (7/23) and 18.8% (9/48), respectively, among patients who did not receive LVA, while it was 4% and 0%, respectively, among patients who received LVA [16,26]. All the other non-randomized case–control studies showed a rate even higher of lymphedema among the controls, going from 8% (19) up to 68%, while among the cases, the rate of lymphedema was significantly lower, from 0% in Yoon’s study, up to 16% in Ozmen’s study. The follow-up rate period was 14.5 months on average [19,26,27].

The relation between the LVA shunting technique used and the rate of cases with lymphedema is worthy of note. The lowest rate appeared in Yoon’s study in which end-to-end LVA was performed on 21 patients with a lymphedema rate of 0% within 6 months of follow-up, while the highest rate appeared in Ozmen’s study, in which, among 110 patients treated with a simplified version of the LYMPHA technique, 18 developed lymphedema (16%) within 84 months.

Despite the lower rate of lymphedema registered in all the mentioned papers, skepticism has emerged over the years of the cost incurred of an additional procedure that requires microsurgical expertise. To solve this controversy, in 2019, Johnson et al. evaluated the cost–utility of a surgical procedure performed for the prevention of lymphedema in a patient population undergoing ALND or ALND with regional lymph node radiotherapy (RLNR) [38]. Their findings demonstrated that the addition of LYMPHA to ALND and ALND with RLNR was more cost-effective than ALND and ALND with RLNR alone, with favorable cost–utility ratios (ICURs) of $1587.73/quality-adjusted life years (QALY) and $699.48/QALY, respectively. The substantial clinical benefit of LYMPHA easily overcame the cost disadvantage, which is why ICUR in both scenarios had a relatively low amount per QALY. The huge bias of Johnson’s paper, though, was the exclusive analysis of the LYMPHA procedure at the time of ALND. As we have emphasized in this review, the LYMPHA technique is not the only option in the prophylactic surgical BCRL scenario [3].

There are alternative surgical options available, but the efficacy of those conducted concurrently with ALND is still unverified. Following ALND, patients may necessitate RLNR, which in itself poses an independent risk factor for the development of lymphedema [39,40,41]. Currently, there is a lack of published studies examining the long-term patency of anastomoses with respect to the safety of adjuvant radiotherapy. However, in our analysis, a higher incidence of lymphedema cases was observed among patients who underwent RLNR following prophylactic surgery, with 15 out of 253 patients (5.93%) experiencing lymphedema (Table 1 and Table 2). Notably, Pierazzi et al. [24] recently reported the sole series in which LVA was performed distally to the irradiated area after axillary lymphadenectomy and adjuvant radiotherapy, a concept previously suggested by Chen [24,42]. However, the five cases reported in their paper do not allow us to ensure a lower lymphedema rate compared with all the other studies, the patients from which received adjuvant radiotherapy after PLS.

Finally, it is important to mention a recently published randomized control trial regarding the effectiveness of immediate lymphatic reconstruction (ILR) in reducing the occurrence of BCRL following ALND [43]. Using an upgraded version of the LYMPHA technique, Coriddi et al. [43] demonstrated how the occurrence rate of BCRL was 9.5% among those in the ILR group, contrasting with 32% in the control group (*p* = 0.014) [20,43]. A significant limitation in their study, although, is the lack of blinding, as emphasized by the authors themselves. This absence of blinding is linked to the recording of operative details in the operative report, which was easily accessible to the patients. This situation could potentially lead individuals in the control group to be more likely to use compression garments, introducing a potential bias that might impact the final results. Furthermore, the classification of lymphedema based on a 10% relative volume change in the upper arm is viewed as somewhat arbitrary, prompting the suggestion for a more standardized assessment method.

## 5. Conclusions

In recent years, various techniques designed to prevent BCRL have emerged, with ongoing advancements in new strategies. This paper aims to synthesize and analyze the existing literature, intending to offer more robust recommendations regarding the effectiveness of these prevention approaches.

The formulation of strong recommendations in favor of any particular PLS, although, is impeded by low-quality studies marked by significant heterogeneity, short follow-up periods and variability among diagnostic modalities. Nonetheless, a consensus is lacking regarding the preferable PLS technique, and the potential impact of adjuvant radiotherapy on its effectiveness remains unclear.

Our findings suggest that delayed LVA might be considered a standard procedure for the primary prevention of BCRL. However, it is essential to acknowledge the need for high-quality randomized controlled trials to establish evidence-based recommendations in this field.

## Figures and Tables

**Figure 1 jcm-13-00555-f001:**
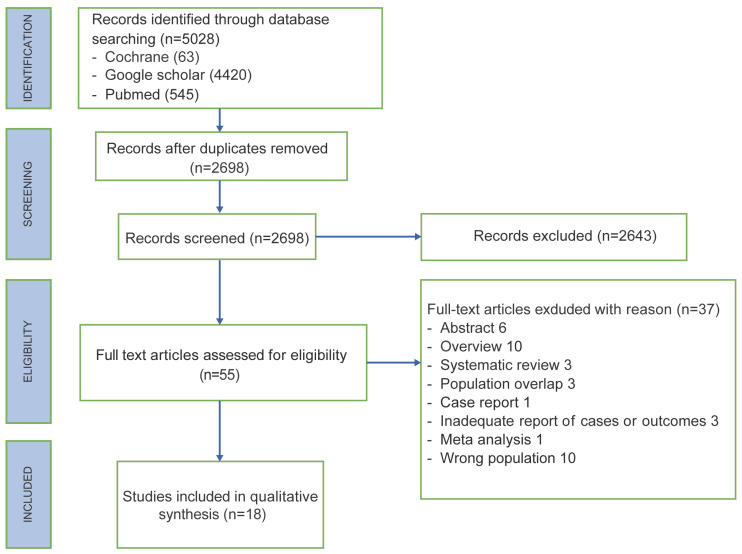
PRISMA flow chart.

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
