# Peer review of "Prevention of Breast Cancer-Related Lymphedema: An Up-to-Date Systematic Review of Different Surgical Approaches"

_jcm, 2024, doi:10.3390/jcm13020555_

Round 1

Reviewer 1 Report

Comments and Suggestions for Authors

It was great that many studies have found, organized, and analyzed the studies well in the areas where many studies are needed in the future and I want to praise the efforts of the authors. However, if you check the following, it will be very helpful to clarify this study.

1. In Figure 1, the number of “Full-text articles excluded with reason” should be 37, but it is indicated as 38, so it needs to be checked and corrected.

2. It would be nice if the LVA feature in Table 1 was also included in the % value as in Table 2.

3. Only one study by Casabona, the second object in Table 2, is missing a publication year, so I think there should be a mark on it.

4. the full term of "LS" written in Table 2 should be added to the abbreviation description below the table.

5. In the reference citation, in the 150th line of the result, please confirm whether references 25 and 30 are correct. The study involving LQOLQ in Table 1 is Hermans 2021, and this study is 18 in reference.

6. In Tables 1 and 2, if the reference numbers are written with the author and the year of publication in the article section, we will be able to identify the cited studies better. ((Example) “Hermans 2021 [18]” )

7. Please check whether reference No. 19 of the 190th row of the Discussion section is an appropriate quotation as the paper that Boccardo and others introduced LYMPHA.

8. When describing the Discussion and Results, it would be nice to include the same topic in one paragraph. As each study describes, the paragraphs are split, so the reader may feel a lot of distraction.

Author Response

Thank you for your comments. You will find the resposnse to them below:

  1. As you correctly pointed out, the correct number of full text articles excluded should be 37
  2. What do you mean with "LVA feature"? I can't find any column under this name in table 1
  3. You are completely right, the publication year should be added
  4. You are completely right, the abbreviation description should be added
  5. References number 25 and 30 are wrong (the article under the reference 25 doesn't involve LQOLQ in its method of lymphedema's evaluation, while the article under the reference 30 has been actually added to the reference list by mistake). Reference number 18 is the right one to be cited
  6. Thank you for the suggestion. It will be done
  7. You are right, the correct citation is the reference under the number 20
  8. Thank you for the suggestion. It will be done

Reviewer 2 Report

Comments and Suggestions for Authors

In the manuscript, "Prevention of Breast Cancer-Related Lymphedema: An Up-To-Date Systematic Review of Different Surgical Approaches," looks to evaluate the current literature on the topic before Aug 22. Immediate lymphatic reconstruction is an ever evolving approach to prevent lymphedema and the studies up to this point have been of lower power with varying follow-up times. This paper looked to summarize this literature to better give recommendations of the efficacy of this technique. The authors did a nice job summarizing the current literature. I think the authors should elaborate on the limitations of the current literature and thus the how it limits the conclusions that can be draw rather then just the sentences 296-298. Additionally, as this study was "Up-to-Date" at the time of submission there has since been a published randomized control trial looking at ILR (Coriddi M, Dayan J, Bloomfield E, McGrath L, Diwan R, Monge J, Gutierrez J, Brown S, Boe L, Mehrara B. Efficacy of Immediate Lymphatic Reconstruction to Decrease Incidence of Breast Cancer-related Lymphedema: Preliminary Results of Randomized Controlled Trial. Ann Surg. 2023 Oct 1;278(4):630-637. doi: 10.1097/SLA.0000000000005952. Epub 2023 Jun 14. PMID: 37314177; PMCID: PMC10527595). I do not necessarily think the study needs to be completely redone but it might be worth noting this as this is an important paper for the topic. Leaving it out in some point would make the Up-To-Date aspect of the paper irrelevant. Other small minor comments:

44 - would also discuss the the possibility of VLNT acting to suck up fluid from the surrounding tissue like a sump pump

227-228 - is this all part of the same paragraph

266 - what is the percentage for Ozmen's study

Author Response

Thank you for your precious comments.

  1. The limitations of our study should be surely clarified more
  2. Aiming our study to be "an up-to-date systematic review", we should speak about the paper you mentioned (Coriddi M, Dayan J, Bloomfield E, McGrath L, Diwan R, Monge J, Gutierrez J, Brown S, Boe L, Mehrara B. Efficacy of Immediate Lymphatic Reconstruction to Decrease Incidence of Breast Cancer-related Lymphedema: Preliminary Results of Randomized Controlled Trial. Ann Surg. 2023 Oct 1;278(4):630-637. doi: 10.1097/SLA.0000000000005952. Epub 2023 Jun 14. PMID: 37314177; PMCID: PMC10527595) in the discussion
  3. adding "the possibility of VLNT acting to suck up fluid from the surrounding tissue like a sump pump" would be redundant since VLNT causes itself the formation of new lymphatic vessels which, over time, will suck up fluid from the surrounding tissue

  4. 227-228 - will be included in the same paragraph

  5. the percentage of Ozmen's study is 16 % and it will be mentioned

Reviewer 3 Report

Comments and Suggestions for Authors

This review article *Prevention Of Breast Cancer-Related Lymphedema: An Up-To-Date Systematic Review Of Different Surgical Approaches*  is clear, well-structured and relevant to this field.

It is difficult to assess the modernity of literature data, since there is no list of references.

Manuscript’s results reproducible based on the details given in the methods section.

The figures and tables in this manuscript correspond to the text and are easy to understand. The information given in the tables is reflected appropriately.

Conclusions consist with the evidence and arguments presented.

Author Response

Thank you for your comments.

The list of references was provided after the submission by mistake.

In attachment the manuscript with the list of references which would need although some modificiations as pointed out by another reviewer.
